# Study on the Applicability of Digital Twins for Home Remote Motor Rehabilitation

**DOI:** 10.3390/s23020911

**Published:** 2023-01-12

**Authors:** Piotr Falkowski, Tomasz Osiak, Julia Wilk, Norbert Prokopiuk, Bazyli Leczkowski, Zbigniew Pilat, Cezary Rzymkowski

**Affiliations:** 1Łukasiewicz Research Network—Industrial Research Institute for Automation and Measurements PIAP, 02-486 Warsaw, Poland; 2Institute of Aeronautics and Applied Mechanics, Faculty of Power and Aeronautical Engineering, Warsaw University of Technology, 00-665 Warszawa, Poland; 3Chair of Clinical Physiotherapy, Faculty of Rehabilitation, The Józef Piłsudski University of Physical Education in Warsaw, 00-809 Warszawa, Poland

**Keywords:** digital twin, exoskeletons, home rehabilitation, human–machine interaction, IoT, motor therapy, robot-aided rehabilitation, remote treatment

## Abstract

The COVID-19 pandemic created the need for telerehabilitation development, while Industry 4.0 brought the key technology. As motor therapy often requires the physical support of a patient’s motion, combining robot-aided workouts with remote control is a promising solution. This may be realised with the use of the device’s digital twin, so as to give it an immersive operation. This paper presents an extensive overview of this technology’s applications within the fields of industry and health. It is followed by the in-depth analysis of needs in rehabilitation based on questionnaire research and bibliography review. As a result of these sections, the original concept of controlling a rehabilitation exoskeleton via its digital twin in the virtual reality is presented. The idea is assessed in terms of benefits and significant challenges regarding its application in real life. The presented aspects prove that it may be potentially used for manual remote kinesiotherapy, combined with the safety systems predicting potentially harmful situations. The concept is universally applicable to rehabilitation robots.

## 1. Introduction

Often, ideas about the future include the ability to control objects over long distances, visiting shops without leaving home, interactive, engaging games or meetings with loved ones anywhere and anytime thanks to a remote connection [1]. Within the last few years, there has been a strong need to bring these closer to the present. The COVID-19 pandemic has proved the need to revitalise the health service and create a new approach to patient treatment during crises. Hospitals, medical centres, and rehabilitation facilities have been crippled, mainly by a shortage of health workers who cannot maintain an adequate level of health services to meet the population’s growing needs. As urbanisation has led to the rapid spread of infectious diseases, while the population is ageing, the number of people requiring constant medical care is increasing [2]. Such challenges are met by telemedicine and telerehabilitation, intensively developed within recent years [3]. With the rise of digital technologies, remote monitoring of patients’ progress in therapy and the assessment of their health has been enabled. This is possible thanks to ECG, blood pressure, and glucose level measurements, among others [4]. Often, rehabilitation of people who cannot sit or stand independently requires the help of up to three physiotherapists, as it is too difficult for one [5]. Digital twins seem to be the solution in such situations. Initially, they were developed as digital equivalents of machines and industrial elements, such as engines or turbines [6]. They may be used to predict the system’s response in critical events and detect previously unidentified problems by comparing the predicted and actual responses [7]. Currently, attempts are being made to introduce this technology into healthcare to forecast potential outcomes of different treatment approaches [8]. Digital twins seem to be an ideal solution for the personalised therapy for each individual with a specific disease or condition [2]. Such models can consider many variables regarding the given case and predict their impact on a patient [9]. Bearing in mind the growing interest in implementing digital twins in medicine, the following paper provides an overview of their use in medicine and industry, an overview of the current needs of physiotherapists and patients during exercise therapy, as well as a discussion of the challenges and the applicability of VR technology for remote movement rehabilitation.

## 2. Motivation and Methodology

The paper aims to present the current uses of the digital twin technology and assess its applicability for home remote motor therapy. The chosen cases are based on the best practices, and not necessarily limited to medicine. They were selected in the process of an in-depth literature overview. Only the most popular and technologically advanced of each type were elaborated on to reduce the number of the presented solutions.

In the process of the literature overview, 64 relevant original papers were analysed. These were selected based on the search within ResearchGate, Google Scholar, PubMed, and IEEE Xplore engines. The results for scientific publications were limited to 2017 and newer. For the search, the following keywords phrases were used: digital twin, digital twin medicine, digital twin surgery, digital twin rehabilitation, digital twin treatment, digital twin industry, digital twin manufacturing, digital twin education, digital twin robotics, digital twin robotics VR, digital twins physiotherapy.

Additionally, an overview of the needs of contemporary rehabilitation is presented. This is intended to form a design intent for the developed technologies for remote home physiotherapy of the future.

The overview of these is based on in-depth interviews with physiotherapists followed by the questionnaire research on a group of 138 professionals and the literature analysis. The investigation method is described in the corresponding section.

In the process of the literature overview, 15 relevant papers were analysed. These were selected based on the search within ResearchGate, Google Scholar, PubMed, and IEEE Xplore engines. The results for scientific publications were limited to 2017 and newer. For the search, the following keywords phrases were used: future rehabilitation, rehabilitation needs, telerehabilitation.

The paper also contains the original concept of remote robot-aided treatment based on the digital twin technology. The presented solution addresses the contemporary problems of unintuitive or ineffective remote treatment. The technology is strongly dependent on the initial state of the art presentation to correspond to actual needs and possible technological advancement. All the discussed ideas can be developed and fulfil the requirements towards home remote kinesiotherapy.

## 3. Overview of Applications of Digital Twins

### 3.1. Healthcare

Medicine is a field that will undoubtedly benefit from the development of digital twins. This is an effect of the rising popularity of smart mobile devices for organised patient data storage, limited human learning capabilities, the need for personalised and targeted treatment, and the need to reduce treatment costs. Introducing the ability to work on models and simulations allows faster, more cost-effective elimination of risk for patients [10]. As described by Volkov [4], digital twins in medicine are typically used for diagnostic purposes. However, their development phase is lean. The innovative applications are formed with new types of sensors or supporting devices combined with the Internet of Things (IoT) technology [4]. Burdea, in his paper from 2003 [11], presented a concept of virtual rehabilitation dedicated to patients considering types of their diseases. However, as with most cases in medicine, the application of VR was limited to presenting a gamified graphical user interface. The presented models reflected the movement of the user’s involved body segments, but there was no force feedback provided. However, such a methodology enabled the enhancing of patients’ engagement.

One of the most significant factors affecting the general population’s health is physical activity. Coaches are now being trained to motivate people who are not interested in sports. However, not everyone can afford a private trainer. Digital Twins can allow highly customised workouts for people who do not collaborate with professionals [12]. Another global health threat is type two diabetes. However, with the help of digital twins, the consequences of this disease may be reduced. The use of digital twins for patients with type two diabetes can decrease systolic blood pressure, heart rate or body mass index. The consequence of this is a reduced amount of blood pressure medication [13]. In surgery, digital twins can be used to practice interventions in a simulator to learn the exact anatomy and avoid unnecessary tissue damage. They also contribute to developing new tools, techniques, and therapies, to limit risks corresponding to the surgeries. In addition, digital twins can be used to visualise the current state of patient organs. This helps doctors better prepare for upcoming surgery and explain health issues to non-professionals [14]. The presented technology may also be used for training resident doctors [15]. In cardiology, digital twins can optimise treatment outcomes based on the collected real-time data, which can be additionally involved for further simulations. These contribute to verifying the drug therapies or medical devices in terms of their fitting to convalescents [9,16].

Additionally, cardiac resynchronisation therapy for patients with unclear electrocardiography (ECG) was developed within mechanistic modelling. Based on simulations of the human circulatory system, it was found that the response to cardiac resynchronisation therapy can be predicted from the presence of non-electrical substrates. Mechanistic modelling helped to solve the problem of ablation of ventricular tachycardia. The synergy of models is a unique solution for interpreting structure–function relationships and improving risk assessment in inherited diseases such as hypertrophic cardiomyopathy [17]. In addition to ECG data, electroencephalography (EEG) motor imagery signals can be used to build digital twins. With these data, the system could detect a patient’s motion intention and help make a move using wearable, including soft robots or rigid exoskeletons [14]. Single-cell twinning is an essential link to understanding the metabolic complexity of the human body, particularly considering cancer detection [18].

Moreover, works on the digital twins for multiple sclerosis are promising. They may accelerate advances in diagnosis, treatment, individualised clinical decisions, and more efficient management within this field [10]. The use of digital medical twins does not necessarily refer to the human body. Their use for operating the healthcare systems is equally important. Such applications enable the monitoring of the treatment process, for example. For this, the pre-hospital systems are used for real-time information collection about the trauma, position of the ambulance or even smart devices data on rescuers’ operations. Then, the hospital systems collect and store patients’ data, for example, the vital signs monitored [16,19]. Additionally, ventilation systems are developed to limit the spread of microorganisms based on the predictions of their computational models [20]. Analogically, other medical devices are enhanced thanks to their digital twins. With this technology, the efficiency and safety of assembly are constantly improved [21]. The review of exemplary medical applications of virtual twins is presented in Table 1, with a particular focus on the measured aspects. As can be observed, most of the applications are presented as concepts only. The significantly problematic aspects of their real-life implementation are medical regulations regarding safety and personal data collection [14,16,17].

### 3.2. Automation and Robotics

Despite the rising interest in digital twins for healthcare, they are mainly used in industry. Their application is one of the fourth industrial revolution’s (Industry 4.0) central pillars. Such technology is primarily involved in Product Life Management (PLM) or Smart Manufacturing; however, these are not the only possible applications. Digital twins allow online visualisation of manufacturing devices or robotised workstations. This, in turn, allows engineers to supervise industrial processes from any place on the Earth, which is especially important during an ongoing COVID-19 pandemic [14,22]. In addition, digital twins for monitoring purposes could collect data about the operating machinery. Based on this, the deep learning algorithms could predict mechanical part conditions [14,23].

However, industrial digital twins are not only used to monitor processes. Thanks to augmented reality (AR) technology, such systems allow the collaboration of humans with manufacturing devices, robots or production lines, among others. In the case of production lines, the AR sets display instructions needed to perform a given process and allow the monitoring of its course in an intuitive environment. The presented content is strictly related to the data gathered from the real-life machine and implemented by its digital twin estimating the current operation states [24,25]. On the other hand, the most critical aspect for human–robot collaboration is supervising the manipulator’s work zones. Based on the gathered signals and the digital twin predicting further dynamics of the device, the station’s operator may safely share space with the robotic arm [26].

In addition to AR, industrial digital twins are built within virtual reality (VR). They enable the control and programming of robots in the immersive digital environment. This is superior for complex industrial processes with complicated, precise robot movements, difficult to reach manually with the operator’s panel. The solutions to such challenges are systems reconfiguring robots while manipulating their virtual models [27]. Additionally, embedded safety modules help avoid collisions with the modelled real-life environment [28].

Industrial digital twins are also involved in simulations of production lines’ operations. Such systems are often used as proof of concept at the initial design phase. Their application helps to improve construction and test the solution at various stages—before purchasing components or final launch, among others. Thanks to these, it is possible to create the solution corresponding to the needs without the risk of budget outspending, damaging the devices or injuring humans [29]. The review of exemplary applications of virtual twins related to automation and robotics is presented in Table 2, with a particular focus on the measured aspects. Compared with the medical usage, there are many more examples of real-life application of the digital twins in robotics and automation. However, most are dedicated to visualization and remote programming of the setups.

## 4. Overview of Needs in Rehabilitation

Taking on the topic of telerehabilitation was preceded with three in-depth interviews and the questionnaire research on the group of 138 Polish physiotherapists—mainly specialists in orthopaedics (53.6%; the respondents could select up to three answers depending on the types of diseases they are regularly working with every day), neurology (32.6%), ambulatory rehabilitation (29.7%), sports rehabilitation (28.3%), paediatrics (13.8%), cardiology (8%), and geriatrics (5.8%). The demographic characteristic of respondents are presented in Table 3.

For the purpose of defining needs in contemporary rehabilitation, the questions related to home rehabilitation were analysed. As 38.4% of physiotherapists declared they perform treatment at the patient’s place of residence, the results were considered relevant. Remote treatment can reduce costs for healthcare providers while lowering costs for patients [30,31], especially those with chronic conditions, as it significantly facilitates the process of monitoring their health. It also provides an opportunity for medical consultations for patients in remote locations who cannot quickly and safely reach a doctor [4]. Within the presented research, the investigation was focused on the answers to the following:What are your major problems while performing the therapy? (up to three answers chosen from the list or optionally added and then grouped into the categories)What are your wishes related to physiotherapy, even unrealistic? (up to three answers grouped into the categories)

The results of the mentioned questions are illustrated in Figure 1 and Figure 2. As may be observed, the significant difficulties in kinesiotherapy are connected either with a reduction of physiotherapists’ efforts (physical loads and time consumption), encouraging patients (keeping their attention on the constant level, risks reduction or overcoming anxiety) or therapy planning (adjustment of the proper treatment and measurable assessment of progress). Moreover, most of the respondents pointed at yet unknown State of the Art as the most beneficial for the physiotherapy of the future. At the same time, 26.8% of them have chosen instant transportation to the patient’s house as the answer to the second question.

This analysis was followed by validating the theses for the Polish population regarding global requirements based on the available literature sources. In search of rehabilitation necessities, Kamenov found a great need for an investigation, especially in low- and middle-income regions. The paper elaborates on studies conducted in Mozambique, Malawi, Zimbabwe, Zambia and Lesotho. These show that up to two thirds of people with disabilities living in the mentioned regions who require rehabilitation do not have access to it. Furthermore, there is a growing demand for rehabilitation worldwide due to an ageing population and a more frequent incidence of non-communicable diseases, especially in developed countries. The analysis also identifies numerous barriers to accessing rehabilitation services [30].

Studies indicate that, even in developed countries, access to rehabilitation in rural areas is limited. It is difficult for low-income individuals to receive physiotherapy due to a lack of transportation, poor equipment in the closest facilities, physically inaccessible locations of centres, or high treatment costs. Because of such barriers, distant rehabilitation is attracting increasing interest. Moreover, the cultural aspect is essential while working with patients, which may cause limitations in the work of some specialists (e.g., the need for a same-gender physician as the patient) [31]. Volkov points to the increase in demand for medical services due to the rise in population and life expectancy. Additionally, according to the paper, human-performed therapy is difficult to monitor. Moreover, an increasing demand for medical services and rising prices hinders chronically ill people from affording appropriate treatment [4].

These days, motor rehabilitation is often focused on function-oriented treatment. Physiotherapists concentrate not only on examining values such as muscle strength or range of motion but also on the overall functional state of the patient. Such therapy aims to help the patient fully recover or get close to the functional state before the disease. It is essential for patients who need to perform certain professional or daily-life motor activities [32]. The function-oriented therapy requires the evaluation of the patients’ performance with functional tests. With such tests, the effectiveness of the therapy may be assessed while keeping motivation on a constant level. For this purpose, various examination methods and scales are used, e.g., 6-min walking test [33] or Barthel Index [34].

## 5. Original Concept

### 5.1. Robot-Aided Rehabilitation

The demand for physical support of older physiotherapists and preparing methodology of remote motor treatment led to the need for the overtaking of tiring activities by machines. Hence, an operator-less robotic solution must be implemented for home-based kinesiotherapy. However, current commercial devices are typically either too simple or too complex for self-in-house workouts. The first type of these machines is the one that mechanically supports a small number of DOFs or triggers the motion by the end effector attached to the body in series only [35,36]. With such an approach, a comprehensive treatment of patients with serious diseases may be given, and neither is monitoring their kinematics parameters possible [37,38]. An insufficient number of DOFs do not support tasks with genuine limb motion patterns. For example, a mechanism with three DOFs cannot be used for lifting a glass along a natural trajectory.

The complex rehabilitation robots are typically too large, heavy, and expensive to be purchased and applied by single users [5]. Moreover, they typically require operational space expanding beyond that available in most flats [39]. Hence, they are not applicable for home use, even though their advanced automation systems allow the generation of customised workout routines and the monitoring of performance continuously [40].

According to the aspects presented beforehand, the device for remote home motor therapy should be relatively simple in terms of mechanics, while remaining technologically advanced. Its structure has to enable easy putting on and training within the limited space. Moreover, it must enable control over the dynamic parameters of rehabilitated body parts—either by monitoring corresponding parameters in all the DOFs of machines or by involving an additional bio-signals tracking system. For these reasons, the exoskeleton lightweight structures are considered. They allow direct mobilisation of particular DOFs. Therefore, kinematics of corresponding joints may be calculated based on the data from encoders and impedance-based torque estimations [41].

At present, multiple various designs of exoskeletons for all the body parts have been developed [42,43,44]. As an addition, the ongoing research is focused on reducing the exoskeleton’s mass by computational optimisation and the application of new materials or manufacturing methods [45].

### 5.2. Remote Treatment

With the development of Information and Communication Technologies (ICT), new potential was unlocked for motor treatment. So-called telerehabilitation is a process of clinical therapy provided remotely using ICT [46]. However, so far, it has been mainly focused on non-robotised applications. An example of such an approach is implementing serious games based on different Virtual Reality scenarios. This allows a patient with motor difficulties to perform highly interactive and non-intrusive exercises. Furthermore, such interactive tools could also contribute to the automation of training assessments [47], which may be comparably effective to the co-located method [48]. Nevertheless, the described treatment may be introduced only to patients with minor injuries or diseases, as they need to move independently.

As derived from Section 4, providing distant treatment at a patient’s place of residence is the critical aspect of future physiotherapy. Moreover, this should be complementary to the robot-aided treatment. Hence, it is crucial to develop a stable and efficient method of remote kinesiotherapy using a mechatronic rehabilitation system.

Telerehabilitation can be synchronous (connecting patient and therapist in real-time via dedicated devices), asynchronous (computer-based interventions, which are remotely monitored and adapted offline by a therapist) or a combination of both [49]. The considered exoskeleton should enable onsite and remote treatment. The desired trajectories of the rehabilitated segments should be either programmed numerically or registered with the healthy body parts [39]. Moreover, a list of the standard exercises self-adjusting for the anatomical characteristics of a user should be provided within the system. A physiotherapist should be able to connect remotely and select the settings for the workout routine or use the automated database and optionally modify the suggested set. As for asynchronous telerehabilitation, they should be able to monitor patients’ performance with the minimum latency and possibly react to dangerous situations.

During the session, the robot should either support or resist the intended motion [50]. Some commercial companies investigate even more advanced methods, such as error enhancement [51]. A patient is supposed to act in order to follow the desired trajectory. With the automated settings, the therapy could be realised without the active intervention of a professional, who could supervise multiple persons simultaneously.

However, sometimes the treatment requires unpredictable means. Continuous non-anatomical motion patterns resulting in unequal loads distribution within the musculoskeletal system and extensive efforts [52], or the diseases affecting the neurological system [53,54,55], may create the need for another approach to treatment. In such a case, temporarily leading the body segments manually to find the optimal rehabilitation paths can be inevitable. The synchronous telerehabilitation with the robot could contribute to this. Nevertheless, such an application requires developing a control methodology to enable intuitive programming of the device’s motion mobilising patient regarding anatomical patterns.

### 5.3. Digital Twin in VR Based Control

The concept of the presented solution includes the industrial-like application of the digital twin to robot-aided motor therapy. The rehabilitation robots currently used for kinesiotherapy support or resist the motion of a patient. They may be either attached to the body segments in series (by end-effectors) or parallel (as exoskeletons) [56,57]. They are mainly operated by regulators of the motors controlling them so as to follow the pre-programmed trajectories within the time and under unknown loads. This brings higher accuracy and repeatability than the manual treatment with a physiotherapist, but does not give flexibility within the workouts. On the contrary, manual therapy may bring superior outcomes during specific cases which require real-time modification of exercises or experimental search of the optimal motion pattern [57]. Hence, the combination of both is the most effective when dealing with complex diseases, especially as both may be superior in particular cases [36,58].

While performing the physiotherapy in a clinic, it may be realised by combining manual therapy sessions and training with rehabilitation robots [59]. The real challenge is to transfer this approach to distant therapy.

The presented idea considers using the exoskeleton in the home of a patient connected with the remote control module. The treatment is initially based on the automatically generated routines based on the registered performance. However, all the adjustments may be modified online by the physiotherapist. Moreover, they have an opportunity to constantly monitor patients’ activity based on the impedance of the mechatronic device or additional sensors (e.g., EMG or EEG tracking).

The rehabilitation system is complemented with its digital twin. Additionally, the model contains schematic geometry of the rehabilitated body segment. The digital twin may be displayed in the VR goggles of the physiotherapist, as this provides the best immersion. AR or MR technologies are not considered because the patient is treated remotely, and the real-life environment of the physiotherapist is not connected with the rehabilitation process. Thanks to this, the professional can monitor the actual activity of the patient not only via registered signals but also based on the observed multibody model. As for the manual remote therapy, the physiotherapist should be able to drag the body segments to the desired configurations. This is realised by control over the virtual multibody model. Thanks to the kinematics formulas, the motion of the body segment may be recalculated into the desired motion of the motors. With the presented approach, manual therapy may be performed remotely in an intuitive and immersive way.

The concept of such an application is visualised in Figure 3. The presented method is not limited only to extremities and may be scalable even to the whole body or a group of patients, not necessarily located in the same room.

## 6. Discussion

### 6.1. Benefits

Using technology such as digital twins in rehabilitation will contribute to its cost reduction, as one therapist will be able to provide treatment to multiple patients simultaneously. It will bring precise and individualised physiotherapy to less affluent patients. Moreover, the time spent on related frequent journeys of convalescents or professionals will also be shortened. With a patient-tailored therapy, its time may be decreased as well, cutting costs even further. 

Collecting data from a broad selection of sensors will allow collecting and transferring patient information between specialists. With such capabilities of a system, the interdisciplinary teams could provide holistic care for patients even with rare diseases. Thanks to the distant treatment, barriers such as different locations will be mitigated. This can be crucial for individuals whose access to professional health care is restricted by long distances from specialist centres. Reducing journey times to the minimum while providing common access to knowledge about the patient’s conditions results in the possibility of continuous rehabilitation regardless of external factors. It also reduces the occupancy rate in hospitals, as patients do not have to visit these. Moreover, telerehabilitation may also positively affect the lives of the seriously ill patients’ relatives, who often accompany their kin [31].

The digital twins are an opportunity for those living in underdeveloped areas. They allow health checks by specialists not accessible within close reach. Additionally, their application enables the faster detection of disorders and their further observation. In some cases, this is the only way to monitor people’s health in less developed regions [31]. Nevertheless, making a technologically advanced solution commonly available requires the telerehabilitation systems to remain relatively inexpensive. Thanks to this, hospitals in less developed countries or even single users could afford medical equipment to be widely implemented.

One of the most significant benefits of using digital twins in rehabilitation is the accurate assessment and monitoring of patients’ progress during workouts. Physiotherapists often do not have the tools to evaluate the performance reliably. They use sight observations, goniometers, scales and other instruments that are too inaccurate for deep biomechanical analysis. In addition, physiotherapists are trained mainly for functional diagnoses. On the contrary, the digital twins give insight into dynamic parameters of rehabilitated body segments and time spans of motion trajectories. Hence, this may impact the development of computer-aided physiotherapy and its effectiveness [30].

The databases collected by such systems could become a breakthrough in individual rehabilitation. These may be used to investigate the ongoing efficacy of physiotherapy, assess the patient’s conditions, and automatise the treatment planning. In addition, gathered information may contribute to creating artificial intelligence algorithms for predictive control algorithms or adaptive training [60].

The implementation of digital twin technology within robot-aided treatment can also affect the consistency of procedures in medicine. This includes combining therapy at different stages, from initial intervention, through diagnosis to treatment and rehabilitation. The continuity of the entire treatment process shall benefit from faster recovery and reduced complications [61].

The technology of digital twins is developing rapidly and finding new applications in synergy with other fields [62]. This trend would hopefully apply to its use for medical purposes. Thus, numerous spin-offs of the presented concept are expected. The development of telerehabilitation with a digital twin may contribute to further research into the cyber-physical twins of physiotherapists, reflecting precise extremities’ motion in distant locations. This also raises hopes for new approaches to performing remote surgeries. Along with that, it is also speculated that creating a human digital twin [14] could lead to a revolution in how such operations are planned and performed.

### 6.2. Major Challenges

Undoubtedly, such an innovative approach requires facing and finding answers to significant challenges. As digital twins are informatics constructs, they are exposed to hacking attacks. This is associated with the risk of unauthorised entities taking control over a device collaborating with humans [14]. Moreover, the system is also endangered by the unintended adding or removing of information, by manipulating content, leaking sensitive patient data, or other IT problems such as difficulties with data visualisation [2]. Providers of such solutions have to consider means for personal data protection, granting access to particular functions, and protecting against cyber threats. The approach to patient data storage and use should be transparently specified [2]. Even though robots collaborating with humans have rigorous safety requirements, they face similar problems as industrial devices in terms of cyberattacks. Thankfully, hardware for remote connection with the safety protocols is already commercially available, and the ongoing research on improving the security of the access is being conducted [63].

The solution presented in the paper is intended for people with limited mobility or difficult access to medical centres. Such conditions often go hand in hand with poor access to the Internet. It is not necessarily just a matter of the individual financial situation, but also a specific location, especially for third-world countries [8]. There is also rising concern about the emergence of the social division phenomenon according to possession of technological devices. With the introduction of innovative machines for individual use, this may become a measure of patients’ social status or segmentation according to their medical conditions [2]. Thus, clear legal regulations are necessary, preferably uniformly among countries that decide to use the discussed technology in the field of medicine and rehabilitation. However, such procedures are difficult to set [8].

Currently used mainly in industry, the technology itself should be adapted to medicine, which requires a more precise approach [11]. The accuracy of hand tracking primarily drives accuracy in virtual reality-based control. Commercial VR headsets provide position tracking with a resolution of a few millimetres [64], which, depending on the application, may be sufficient or force the use of more accurate, dedicated systems.

Apart from positional inaccuracies, the control system should also be robust, particularly to the time shifts. Tracking latency introduces a delay of around 20-40ms into the mechatronic system [65]. Additional delay may result from connecting the device with its virtual twin over the Internet. For non-medical applications, technologies with acceptable latencies have been developed [66]. Moreover, solutions for further reduction of the connection lags were also created [67]. Beyond that, the latency can be minimised thanks to using emerging communication technologies such as 5G and wireless communication [68].

Physiotherapists may also encounter problems with the implementation of the presented technology. Even though the solution guarantees high immersion, it has particular limitations regarding control over the device. The physiotherapist would have to undergo thorough training in using the software and learn how to perform remote manual treatment naturally. The required hand motions may differ from the ones typical for conventional therapy. Such a significant shift in working routine may not be easy to implement, especially for physiotherapists with long work experience [11]. To skilfully adjust the treatment in each case, the operator has to know the technology at a deep level [4]. Moreover, the medical profession may be afraid of replacing humans with robots [11]. Furthermore, the patients will have to get used to a lack of interpersonal contact with the therapist, which is an element of the treatment as well [8,69]. Moreover, despite the engaging games and exercises, patients may potentially perform exercises less accurately and intensively without constant onsite supervision [11]. This is particularly risky in the home environment, where they are exposed to all kinds of distractions [69].

In addition, the patients’ safety is a crucial aspect during treatment. Therefore, a system of controlling the device should react to the user’s discomfort to prevent the hazard of injuries. This is particularly important for patients with comorbidities, especially spasticity [70]. As hypertonia is relatively quick and may even result in muscle damage [71], the operation of the device should be complemented with constant safety monitoring processes.

The difficulties are also related to the sensory perception during teleoperations. Conventional manual therapy performed onsite involves the physiotherapist’s physical engagement. Thus, the specialist has haptic feedback, used to detect irregular organism behaviour. Such feedback is based on kinaesthetic and tactile information [72]. The first one includes reaction forces occurring in the tissues of extremities used for mobilization, while the second relies on neuroreceptors and allows sensing of the touch surface characteristic. Lack of this perception during remote manipulation of the body segments may force potentially dangerous configurations as an effect of having no resistance control. Therefore, both software and mechanical safety systems should be implemented. Initially, the ranges of the device’s reach should be limited according to the patient’s capabilities [73]. Then, the control algorithm has to be enriched with predicting and overcoming hazardous configurations. It may be based on impedance-force control and additional bio-signals tracking systems [74], as their convolution may cause characteristic patterns to be detected with artificial neural networks [12]. Moreover, implementing AI-based control algorithms can also be used to forecast the dynamics of the rehabilitation system and exclude the risk of exceeding acceptable ranges of motion for an individual patient [75]. Thanks to this, the real-life time of sensor data collection will have less impact on safety.

## 7. Summary

The presented State of the Art proves that the technological advancement in the field of digital twins corresponds to the current needs for rehabilitation and enables remote robot-aided therapy. Combining VR-based control with mechatronic kinesiotherapy devices gives an opportunity to drag patients’ body segments during conventional treatment. Thus, this allows treatment individualization and on-spot modifications during unexpected problems with motion patterns. Thanks to such an approach and the automation of the process, faster improvements in the users’ physical conditions are expected.

Such an approach enables the overcoming of the main problems of physiotherapists and patients. These are, as investigated, mainly correlated with a measurable assessment of performance and decreasing time spent on commuting between specialist centres and private houses. Implementing digital twins for robot-aided kinesiotherapy additionally allows international therapy, where the most experienced specialists can work with difficult cases. Application of this may bring particular benefits to people living in rural areas, severely immobile patients and the elderly, in general. This remains in line with the Sustainable Development Goal 3 of the United Nations. Hence, it has a large-scale worldwide impact.

Major challenges towards the presented concept include providing stable Internet connection in less-developed regions, resulting in lags of the system, cybersecurity, precision of the device operated with VR sets, and user safety. The last aspect creates the need to design a system that continuously monitors users’ performance and predicts hazardous configurations of the exoskeleton. With this, the described solution should be capable of remote use.

As for the continuation of the research, a digital twin of the ExoReha rehabilitation exoskeleton developed in the Industrial Research Institute for Automation and Measurements PIAP, based on the presented methodology, will be developed. 

## Figures and Tables

**Figure 1 sensors-23-00911-f001:**
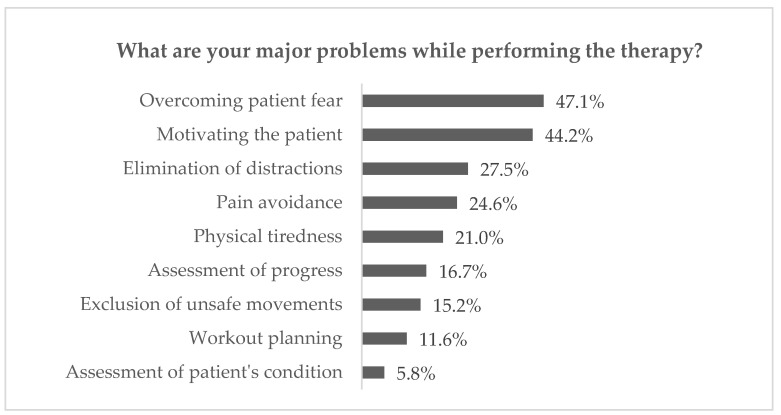
Answers to question 1 from questionnaire research.

**Figure 2 sensors-23-00911-f002:**
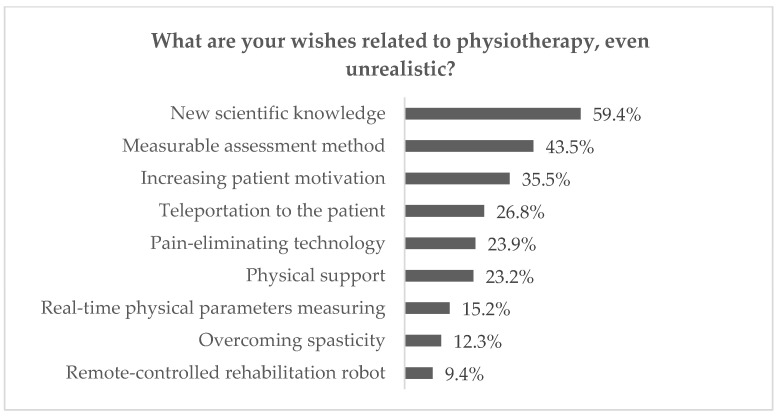
Answers to question 2 from questionnaire research.

**Figure 3 sensors-23-00911-f003:**
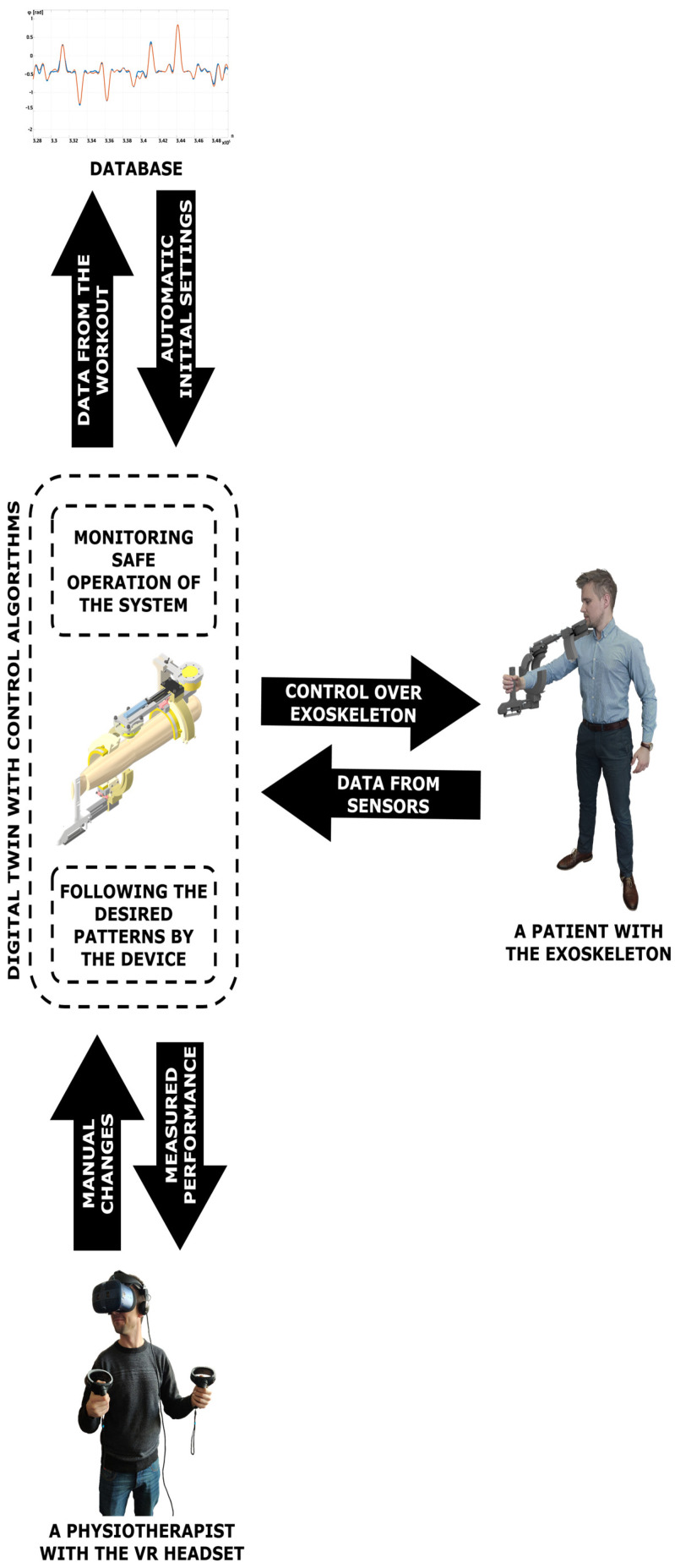
Block scheme of the system for digital twin based remote motor rehabilitation.

**Table 1 sensors-23-00911-t001:** Review of medical applications of virtual twins.

Application	Measured	Reference
Precision treatment with the prediction of future health states	Heart rate, blood pressure, blood glucose, blood BHB level, weight, sleep parameters, step count	[13]
Full lifecycle management (concept presented only)	Heart rate, blood pressure, blood glucose, weight, respiration, exercise volume, emotional changes	[14]
Trauma management (integration of multiple digital twins—concept presented only)	GPS-based position of the rescue vehicle, medical parameters measured in a hospital or by medical rescuers	[16]
Cardiovascular medicine (concept presented only)	Information from private wearable sensors of patient	[17]
Cancer preclinical investigation	Metabolic parameters of cells	[18]
Trauma management	Vision of the trauma team members	[19]

**Table 2 sensors-23-00911-t002:** Review of applications of virtual twins related to automation and robotics.

Application	Measured	Reference
Prediction of the machining tool condition	Forces, vibrations, acoustic emission	[23]
Visualisation of data regarding industrial process	Devices’ process-related parameters	[24]
Visualisation of the assembly process	Vision of the operator	[25]
Safety monitoring for human–robot collaboration	Human kinematic parameters, robot’s kinematic parameters	[26]
Immersive remote programming of industrial robots	Robot’s kinematic parameters, movements of the VR controllers	[27]
Immersive remote programming of industrial robots	Robot’s kinematic parameters	[28]

**Table 3 sensors-23-00911-t003:** Demographic Characteristic of Respondents.

Age [years]	Below 25	9.4%
	25–30	40.6%
	31–40	41.6%
	41–50	5.8%
	Over 50	2.9%
Size of the city where the job is performed [number of inhabitants]	Over 250,000	36.2%
	100,000–250,000	15.2%
	50,000–100,000	12.3%
	10,000–50,000	19.6%
	5000–10,000	8.7%
	Below 5000	8%

## Data Availability

Data sharing not applicable.

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
