# Peer review of "Study on the Applicability of Digital Twins for Home Remote Motor Rehabilitation"

_sensors, 2023, doi:10.3390/s23020911_

Round 1

Reviewer 1 Report

- Thorough revision for English language is required

- Fill in the huge empty space in page 8

- I expected a more detailed summary; given the vast ideas discussed in the survey, the summary is too short and too conclusive. I expect it to have detailed points and research directions where researchers can make good use of.

Author Response

Thank you for your quick and professional review. The paper was corrected according to all your suggestions:

  1. A professional proofreader realised the revision of English.
  2. The empty space is filled.
  3. A more detailed summary with a special focus on possible application and impact is added.

Reviewer 2 Report

Reviewer Response

This review aimed to explore the concept of ‘Study on applicability of digital twins for home remote motor rehabilitation’. This topic could be also interesting for readers; however, several unclear issues should be addressed by authors previously to be considered the manuscript for publication. The followings are few of my comments.  

1.      For line 176~178, mainly specialists in orthopaedics (53.6%), neurology (32.6%), ambulatory rehabilitation (29.7%), sports rehabilitation (28.3%), paediatrics (13.8%), cardiology (8%), and geriatrics (5.8%).

What were the specialist characteristic of respondents, if there were overlapping? Why the percentage summation was not 100% ?   

2.      For line 199~200, At the same time, 26.6% of them has chosen instant transportation to the patient’s house as the answer for the second question. Which was not consistent with Figure 2. Answers to question 2 from questionnaire research, Teleportation to the patient 26.8%. 

3.      For the concept of such an application is visualized in Figure 3. Why wearable virtue reality device was schemed and displayed in the physiotherapist by the VR goggles? How the physiotherapist controlled over the virtual multibody model in an intuitive way? If there were any superiorities then AR/MR or just via computer screen for online monitoring by physiotherapist?  

4.      How to improve the accuracy of the simulation, the real-time of sensor data collection and the AI model calculation to ensure the quality of the clinical  Digital Twins?

5.      Need to correct

line 53   nology for remote movement rehabilitation.

line 174  4. Overview of Needs in Rehabilitation

Author Response

Thank you for your quick and professional review. The paper was corrected according to all your suggestions:

  1. The respondents could select up to three answers depending on the types of diseases they are regularly working with every day – this sentence is added to the article
  2. There was a typing mistake – it is corrected now (should be 26.8%)
  3. The reason for VR selection was added to the description of Figure 3.
  4. The section dedicated to challenges was completed according to the question.
  5. The mentioned errors are corrected.

Reviewer 3 Report

This is a review paper on digital twins in rehabilitation. The presented State of the Art proves that technological advancements in digital twins satisfy the current rehabilitation needs and enable remote robot-aided therapy.

Review papers should, however, follow other review paper formats. Creating a table listing the reference articles for this paper, for example.

The paper recruits several diseases to demonstrate the applicability of digital twins for home remote motor rehabilitation. Rather than focusing on individual diseases, it would be better to concentrate on group diseases. You can get information about orthopedics, neurology, ambulatory rehabilitation, sports rehabilitation, pediatrics, cardiology, and geriatrics. The authors might consider focusing on orthopedics (53.6%) to find more evidence to explain the phenomenon.

Author Response

Thank you for your quick and professional review. The paper was corrected according to all your suggestions:

  1. The tables are added as for this article published in Sensors: https://www.mdpi.com/1424-8220/20/3/592?type=check_update&version=2
  2. We have focused mainly on orthopaedics and neurological diseases that often come together. I have presented the cases more accurately in the tables in line with the suggestion about describing the phenomenon regarding the most popular groups.